# Effects of Competitive ELISA-Positive Results of Piroplasmosis on the Performance of Endurance Horses

**DOI:** 10.3390/ani12050637

**Published:** 2022-03-03

**Authors:** Daniel Bravo-Barriga, Francisco J. Serrano-Aguilera, Rafael Barrasa-Rita, Miguel Ángel Habela, Rafael Barrera Chacón, Luis Javier Ezquerra, María Martín-Cuervo

**Affiliations:** 1Parasitology and Parasitic Diseases, Animal Health Department, Veterinary Faculty, University of Extremadura, 10003 Cáceres, Spain; dbravoparasit@unex.es (D.B.-B.); fserrano@unex.es (F.J.S.-A.); mahabela@unex.es (M.Á.H.); 2Animal Medicine Department, Veterinary Faculty, University of Extremadura, 10003 Cáceres, Spain; rafaelbarrasa@hotmail.com (R.B.-R.); rabacha@unex.es (R.B.C.); ezquerra@unex.es (L.J.E.)

**Keywords:** equine piroplasmosis, ELISA, hematology, *Theileria equi*, performance, endurance

## Abstract

**Simple Summary:**

Equine piroplasmosis (EP) is a common infectious disease in southern Europe. To better understand the impact and influence of EP on the performance of endurance horses, we collected blood samples from national elite horses during different endurance competitions. The horses were tested against piroplasmosis, and several blood parameters related to performance were also evaluated. It seems that horses without clinical signs of piroplasmosis can participate without performance impairment in competitions of up to 80 km. Although it is recommended that longer distance competitions should be further evaluated, this is the first step for decision-making by organizers and participants in this sport.

**Abstract:**

Endurance is an increasingly popular equestrian sport. However, in southern Europe, there is a high prevalence of horses that are asymptomatic carriers of equine piroplasmosis (EP), a tick-borne disease that could affect their performance. This study aimed to evaluate the impact and influence of EP on the performance of endurance horses. Blood samples were collected from 40 horses in Extremadura, Spain, before and after a race, in different national elite horse endurance competitions. Hematological and biochemical parameters and EP seroprevalence were analysed by competitive enzyme-linked immunosorbent assay. The global seroprevalence of EP was 70%, with 27 horses testing positive for *Theileria equi* (67.5%) and three (7.5%) for *Babesia caballi*, with two of these horses (5%) positive for both. Approximately 82.5% of the horses (33 of 40) completed the competition, with no influence on performance or position achieved in those with subclinical parasitosis. There were also no significant differences in hematological or biochemical values between seropositive and seronegative horses. The data suggest that horses without clinical signs of EP can participate without performance impairment in competitions of up to 80 km. Although it is recommended that longer distance competitions should be further evaluated, this is the first step for decision-making by organizers and participants in this sport.

## 1. Introduction

Endurance is an equestrian discipline that has been expanding in recent years. It has been recognized and regulated by the *Fédération Équestrian Internationale* (FEI) since 1982, and it consists of running distances of 40, 80, 120, or 160 km [1]. The horses must pass several veterinary controls (vet-gate) that are performed by FEI-authorized veterinarians, or national-authorized veterinarians in national competitions, at the beginning of the race and the end of each stage (generally every 20 km). If a horse does not pass a veterinary inspection, it is automatically eliminated from the competition. The veterinary inspections include a complete clinical examination: heart rate, mucous membranes (colour and moisture), dehydration (measured as the time it takes for a pinched skin fold over the point of the shoulder to flatten), gut sounds, muscle condition, and regularity of gait (evaluated in trot). Horses may be disqualified due to lameness or metabolic problems. In recent years, there has been an increase in the number of studies aiming to understand the physiological mechanisms that occur in horses participating in endurance [2,3,4], as well as the risk factors that may be involved in the elimination of the competitions [5,6,7,8,9,10,11]. However, to the best of our knowledge, there are no studies on the influence of EP on these horses.

Equine piroplasmosis is a tick-borne disease caused by the obligatory intraerythrocytic protozoa *Theileria equi (T. equi)* and *Babesia cabali (B. caballi)* worldwide [12]. Six genera have been implicated as competent vectors for *B. caballi*, *T. equi*, or both: *Amblyomma*, *Dermacentor*, *Ixodes*, *Haemaphysalis*, *Hyalomma*, and *Rhipicephalus* [13]. It has been reported in many European countries, including Spain, with high seroprevalences described by numerous authors [14,15]. Infection occurs with different degrees of hemolytic anemia, multiorgan failure, and associated systemic signs [16], manifesting in equids as peracute, acute, subacute, or chronic forms [17]. In the case of chronic infection with *T. equi* and/or *B. caballi*, the disease can manifest with non-specific signs such as lethargy, partial anorexia, jaundice, weight loss, and poor performance [18,19]. In addition, EP has a great economic impact on the exportation of the horse industry, because the majority of destination countries require antibody analysis [20].

Therefore, the increasing participation and intensification of these competitions and the high prevalence of asymptomatic carriers of EP in this part of Europe indicate the need for further research on this topic. We aimed to conduct the first prospective study to assess the impact of piroplasmosis on the athletic performance of endurance horses and blood parameters, and whether these could be related to the elimination of horses during competition.

## 2. Materials and Methods

### 2.1. Study Design

Serological and hematological studies were conducted on horses randomly selected from among the volunteers in different national elite horse endurance competitions (14 equids in 40 km, 11 in 60 km, and 15 in 80 km races) that took place in Extremadura, western Spain, in 2019.

For this study, informed written consent was obtained from the owners, as well as permission from the regional Spanish federation. Additionally, epidemiological data were collected using a brief questionnaire answered by the rider or owner. The questionnaire included the race, sex, age of the horse, distance from the competition, feeding of the animal, and the horse’s previous experience during that year in endurance races.

Horses included in the study were classified by the type of competition, and if they finished the competition (group A), or were retired or excluded from the competition (group B), as defined by the FEI and the National Federation in the endurance rules.

The variables studied under these factors were: (a) serologic status with respect to EP; (b) hematological values at rest before participating in the sporting event and once the horse’s participation ends; (c) beats per minute during veterinary controls; and (d) final position reached by the horses in the competition.

### 2.2. Sample Collection

Two blood samples (10 mL) were collected by venipuncture of the jugular vein using sterile vacuum tubes containing lithium heparin and ethylenediaminetetraacetic acid (EDTA) (BD Vacutainer^®^, Franklin Lakes, NJ, USA, EUA). The first sample was taken after the horses were recorded at veterinary control stations before starting the competition, and the second at the end of the competition, or after they were eliminated in the vet-gate control or removed by the rider due to different causes. All the samples were refrigerated at 4–8 °C. EDTA samples were analysed on the same day to obtain the hematological variables and the heparin tubes were centrifugated for posterior analysis. The tubes were centrifuged at 1000× *g* for 10 min at 4 °C, and the plasma was stored at −40 °C until further biochemical analysis (no more than 1 week). Beats per minute (BPM) values were also recorded in veterinary control (three times in the 40 km competition, four times in the 60 km competition, and five times in the 80 km competition).

### 2.3. Parasitological Methods

A competitive enzyme-linked immunosorbent assay test (cELISA) was performed using a commercial kit following the manufacturer’s instructions (VMRD cELISA test, Inc. Pullman, WA, USA) to detect the antibodies against *T. equi* and *B. caballi*. The cELISA is the test of choice in many countries because of its high sensitivity in cases of unapparent and chronic infections [21]. Optical density was determined at 620 nm using an ELISA reader (Thermo Electron Multiskan Ascent Microplate Reader, Thermo Fisher Scientific, MA, USA). Samples associated with percent inhibition (PI) values >40% were considered positive. Additionally, cytology was performed using blood smears stained with Giemsa for microscopic observation (×1000) of intraerythrocytic piroplasms, in either ring-form (*T. equi*) or piriform (*B. caballi*).

### 2.4. Hematological and Biochemical Analysis

Several hematological and biochemical parameters were determined before and after the competition for comparison. Total protein (TP) in plasma was measured using refractometry, and the results were expressed in g/dL. Plasma concentrations of creatinine kinase (CK), aspartate aminotransferase (AST), and lactate dehydrogenase (LDH) were measured using commercial kits from Spinreact^®^ Laboratories for an automatic blood chemistry analyser (Saturno 100, VetCrony Instruments, Roma, Italy). Hematological parameters, including red blood cells (RBC), hematocrit value (Ht), and white blood cell count (WBC), were determined using an automatic blood analyser (Mindray^®^ BC-5300; Vet Spinreact, Sant Esteve de Bas, Girona, Spain).

### 2.5. Statistical Analysis

Descriptive, graphical, and inferential statistics of the data were performed in programmed dynamic reports using R software version 4.0.5 [22,23] to ensure the reproducibility and traceability of the results.

The prevalence of antibodies by ELISA was estimated from the ratio of positives to the total number of samples, with an exact binomial confidence interval (CI) of 95% based on the scoring method described by [24].

To compare groups or test correlations among variables, non-parametric methods (Wilcoxon signed-rank test for repeated measures, Wilcoxon rank-sum test for non-paired samples, and Kendall rank correlation test) were used because normality of continuous variables was generally rejected with the Shapiro–Wilk test considering the limited sample size. The relationships among variables were also investigated by hierarchical clustering via multiscale bootstraps with the help of the R package pvclust [25,26].

Results were considered statistically significant at *p* < 0.05. Except for groups of results, *p*-values are not reported as statements of inequality because the exact values are more informative of the probability of the data under the null hypothesis [27].

## 3. Results

### 3.1. Equine Piroplasmosis Test

Blood smears of all 40 horses were negative for *T. equi* or *B. caballi*. However, the cELISA results demonstrated 70% (95% CI = 54.57–81.93) of positives to EP. Seroprevalence of IgG antibodies against both species was different (*p* = 1.1 × 10^−7^), being of 67.5% (95% CI = 52.02–79.92) for *T. equi*, and only 7.5% (95% CI = 2.58–19.86) for *B. caballi*, with two of these horses (5%, 95% CI = 1.38–16.50) being positive for both.

Differences in PI in the cELISA between the classified and excluded horses were not significant in any case (*p* = 0.89).

### 3.2. Horse Performance and Endurance

Group A, which finished and classified the competition, was formed by 33 horses, of which 23 (69.7%) were positive for EP.

Seropositive horses to *T. equi* obtained the best position in one race modality and worst in two (Figure 1); however, the general performance was highly similar to that of seronegative horses in races of 40 km (*p* = 0.660), 60 km (*p* = 0.791), and 80 km (*p* = 0.667).

There were no significant differences in the distribution of arrival positions between seronegative and seropositive horses in the Wilcoxon rank-sum test (*p* = 1). The correlation between classification and PI values in the cELISA results was also negligible for *T. equi* (τ = −0.006, *p* = 0.60).

The horses in group B were excluded during competition by veterinarians due to lameness (n = 3), metabolic reasons (n = 1), tachycardia (n = 1), or voluntary retirement by their riders due to lameness once they reached the goal (n = 2). The distances in this group ranged from 20 to 80 km (mean, 48.57 km). Five of these horses were positive for *T. equi*, thereby showing a highly similar seroprevalence (71.4%) to group A, despite being a small group.

Differences in hematological and biochemical variables between groups A and B were not significant in any case. The same was true when comparing all horses with EP and seronegative animals, or only within group A.

In hierarchical clustering (Figure 2), arrival position was closely related to resting BPM before the race within a cluster that aggregated the other BPM controls, hematological variables, and some biochemical variables. The Kendall correlation coefficient (τ = 0.20) suggested that low resting heart rates were related to better performance (lower position number); however, this correlation was not statistically significant (*p* = 0.14). However, in the second vet-gate at 20 km, this correlation was significantly inverted (τ = −0.29, *p* = 0.03). Horse performance also demonstrated a correlation with pre-race CK levels of doubtful significance (τ = 0.25, *p* = 0.0538), which was not suggested by hierarchical clustering.

The evolution of BPM in seronegative and seropositive horses was also similar. The median values were lower in horses with piroplasmosis; however, these differences were not significant in the Wilcoxon test (Figure 3). This is in line with the lack of correlation between the BPM and cELISA results in the Kendall test.

The initial BPM score was significantly correlated with the pre-race RBC count (τ = 0.30, *p* = 0.014) and post-race TP concentration (τ = 0.25, *p* = 0.04), and negatively correlated with the final WBC count (τ = −0.28, *p* = 0.02). The following BPM controls also demonstrated significant or doubtful correlations with both Ht and TP after the race (τ = 0.23, *p* = 0.07; τ = 0.23, *p* = 0.08, and τ = 0.29, *p* = 0.11 for Ht; τ = 0.2, *p* = 0.11; τ = 0.26, *p* = 0.05; and τ = 0.4, *p* = 0.05, respectively). However, these variables that appeared to be related to BMP, as the remaining hematological and biochemical variables, did not correlate with horse classification (*p* > 0.24).

### 3.3. Influence of Piroplasmosis Status in Blood Parameters

Globally, exercise significantly increased all hematological parameters and plasma muscle enzymes owing to hemoconcentration and muscular stress (Table 1).

Considering the average count of each horse, the mean RBC count was almost equal (*p* = 0.434) in seropositive animals (8.2 × 10^12^/L) and seronegative horses (8.3 × 10^12^/L). Similarly, Ht, WBC, and TP concentrations reflected a similar hemoconcentration after the race in both groups.

No significant differences were detected between negative and positive horses with respect to serum enzymes, with the lowest *p*-value for AST before the competition (*p* = 0.126). AST also showed the least significant increase after the endurance challenge (*p* = 0.0018). That is, all hematological and biochemical parameters changed after exercise; however, there was no significant difference between non-infected horses and those serologically positive (Figure 4).

With respect to the cELISA results in PI, the hierarchical cluster analysis aggregated both PIs in the same cluster only with AST measures, CK pre-race levels, and iron post-race levels. However, the results for *T. equi* were not significantly correlated with any variable, with a minimal *p*-value for the final LDH concentration (*p* = 0.099).

## 4. Discussion

To the best of our knowledge, this is the first study to describe the influence of piroplasmosis seropositivity on horse performance in endurance races, particularly in Europe.

The only other seroprevalence study for EP in endurance horses has been reported in Brazil by [28], with 77.3% of participants in line with our findings. However, the influence of piroplasmosis on performance has not been analysed.

Although the blood smears for EP were all negative, we detected a seroprevalence of 70% using ELISA. According to previous studies, this method has very low sensitivity for moderate and mild parasitosis [12]. Therefore, the use of immunological diagnostic techniques [29] is recommended to improve the detection of asymptomatic individuals, such as those included in this study.

Studies conducted in Spain found high values EP seroprevalences ranging from 24.1% to 68.3% [14,15,30,31], with the highest prevalence in the southern regions of Spain. Overall, the true seroprevalence of *T. equi* was significantly higher (*p* < 0.001) than that of *B. caballi*. In addition, the predominance of *T. equi* over *B. caballi* observed in our study has already been observed in Europe [14,15,32,33,34,35,36,37].

The unique variable that grouped ELISA with values in the cluster dendrogram before and after the race was AST. This enzyme has been linked to the centrilobular necrosis of piroplasmosis due to reduced blood flow in the liver [38] and can be elevated by some degree of hemolysis [39]. Some degree of cellular damage is expected in infected horses after extreme exercise. However, the evidence for this relationship was sufficiently small to disregard the null hypothesis.

On the contrary, the clustering of ELISA with CK activity was less clear; however, it was correlated with respect to serum reactivity to *B. caballi* antigens after exercise. As only three horses were positive for medium CK activity, this correlation was primarily due to increased CK activity in negative horses with a relatively high PI. Mid-level elevated CK activity has been found in horses with clinical babesiosis [40,41]; however, to the best of our knowledge, this has not been linked with ELISA results in asymptomatic horses.

Exercise greatly increased the CK values in both groups of horses [42]. This is related to the change in permeability of the cell membrane of musculoskeletal cells due to oxidative peroxidation [43,44]. Therefore, this enzyme is a marker of muscle damage that increases in value after intense/prolonged efforts [45]. Parasitism by *T. equi* also induces muscle–skeletal cell damage by hypoxia, increasing CK concentrations, although this effect has only been observed before competition [3,30]. In our study, seronegative horses had a higher mean concentration of this enzyme; however, this difference was not statistically significant.

Differences with respect to the ELISA results for all remaining variables were not significant. While evidence of hemoconcentration and increase in enzyme levels due to exercise was evident in both infected and non-infected horses, there was no evidence of interaction with antibody levels. Tachycardia during the challenge can be related to erythrocytic and protein concentration parameters, expected due to prolonged exercise; however, neither antibody reactivity nor horse performance can be linked to these ELISA parameters, which could be indicative of dehydration. Training causes an increase in the RBC count, which explains why horses that get better positions in the competition and are better trained to obtain higher values of this parameter [46]. Seropositive horses had lower counts, especially after exercise, in comparison with seronegative horses. The hematocrit of the seronegative horses was also higher before and after the race, but this variation was not significant. Other studies have indicated that changes could be explained by a splenic contraction of infected animals [2,45,46,47], releasing all the blood volume stored in the spleen, although horses that suffer from the disease have a smaller extra volume stored because of the destruction of red blood cells caused by the parasites, even if it is subclinical [12,48]. This lower RBC count and hematocrit in horses seropositive for EP may cause a decrease in sporting performance [49,50], resulting from slight anemia caused by chronic infection.

Leukopenia has also been described in natural piroplasmosis [51]. Leukocyte cell counts were also slightly lower in seropositive animals in our study in pre-and post-race samples (pre-race: seropositive 8.4 ± 1.8 × 10^9^/L and seronegative 8.7 ± 1.8 × 10^9^/L; post-race: seropositive 10.8 ± 2.6 × 10^9^/L and seronegative 11.0 ± 2.6 × 10^9^/L), but again without statistically significant differences with respect to seronegative horses. Moreover, horses with subclinical disease presented leukocyte counts within normal values [16], which is in line with the results described in other studies [45].

There were no statistically significant differences in total protein between the seropositive and seronegative horses. Seronegative horses had a higher total protein average than seropositive horses, especially after the competition. This increase is due to dehydration caused by the loss of extracellular fluid [2,3,46], in addition to an increase in globulins in response to exercise stress [47,52]. Increases in acute phase proteins have also been reported [53]. This parameter is highly stable, with little variation between individuals, and therefore does not provide much information regarding subclinical disease [51], since there was no association between hyperproteinemia and *T. equi* infection in the horses sampled.

There were no significant differences in LDH concentrations between the seronegative and seropositive horses. LDH concentrations increased after exertion in both groups. The increase after exercise is due to damage to musculoskeletal cells and hepatocytes caused by oxidative peroxidation [44], and hypoxia owing to exercise intensity [16].

The subclinical condition of piroplasmosis did not cause statistically significant differences in the AST concentration. Before exercise, the seronegative horses had higher concentrations than the seropositive horses; however, the situation was partially reversed after the competition. Despite this increase, the values fell within the expected range. This enzyme increases as a consequence of energy requirements due to competition effort and as a result of the damage to the muscular cells, in relation to the distance travelled [44,47,54]. AST levels in Brazilian sport horses subjected to exercise peaked 6 h later [55]. Hyperbilirubinemia and increased hepatic enzyme activities (Alkaline Phosphatase (ALP), AST, and Gamma-glutamil transferase (GGT)) frequently present in horses with clinical piroplasmosis, and are attributed to a reduction in blood flow to the liver, which, in severe cases, can lead to centrilobular necrosis. Lesions in the lungs of these horses can also contribute to increased levels of these enzymes [38].

The low count of RBCs and hematocrit at the beginning of physical effort is an indicator of physical fitness and sports performance in endurance horses [47]. These are useful values to measure at the beginning and end of training sessions to monitor the conditioning of the horses during the training period. The hematocrit value and concentration of total plasma proteins, CK, AST, and LDH can help prevent and diagnose early excessive metabolic stress in endurance horses [54]. Muscle injury is indicated by elevated levels of AST and CK [40,56] and can be used to routinely monitor the effects of exercise on athletic horses [57]. Overall, the average concentrations of CK, LDH, and AST were the parameters with the greatest variation, both in mean values and between individuals, whereas the concentration of total plasma proteins was the variable with greater stability between groups. Additionally, the basal values in both groups (with or without subclinical disease) were within the physiological range, and variations after exercise were expected [16,46,54]; therefore, a decrease in performance in the seropositive horses during the competition cannot be deduced.

Despite the variations found, we must consider that all values were within the physiological range in all animals [16]. Moreover, we did not detect a relationship between the position reached at the end of the competition and the presence of EP antibodies. These results are similar to those reported for horses under stress conditions with subclinical parasitosis [58].

In a study in Galicia (Spain), among the non-parasitemic control horses, the presence of antibodies against *T. equi* or *B. caballi* was not associated with significant changes in the hematological or biochemical parameters [30]. However, a future study should be conducted with a follow-up in the days after the competition to observe if the stress induced due to the exercise and displacements causes a relapse of the acute phase of the disease [40].

This work is a first approach to the problem of EP in endurance horses. The limitations of the study are the medium level of the competitions (80 kms) that cannot necessarily be extrapolated to a higher lever (120 to 160 kms). In addition, polymerase chain reaction (PCR) was not performed this time, so some positive cases may have escaped diagnosis. It is recommended to use molecular techniques in future studies to complement immunological techniques.

## 5. Conclusions

We found a high seroprevalence of piroplasmosis in the analysed endurance horses from Extremadura. However, when this parasitosis was subclinical, it did not appear to have a clear impact on sports performance in endurance competitions up to 80 km. Therefore, these results indicate the need to continue studies with this approach, evaluating more horses and in competitions of longer distances (120 km and 160 km) to corroborate these preliminary findings.

## Figures and Tables

**Figure 1 animals-12-00637-f001:**
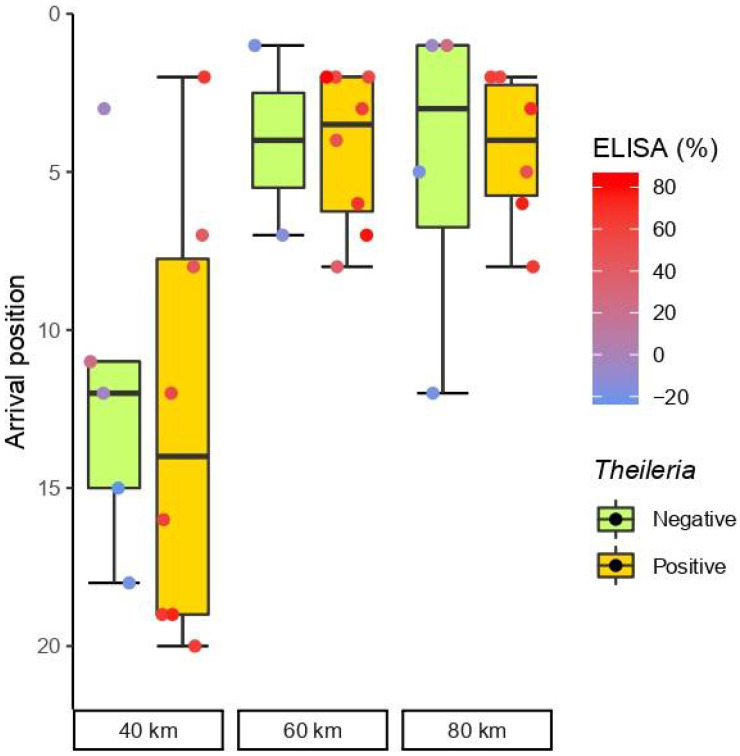
Box and whisker plots of the race position (*y*-axis) of positive and negative horses to *T. equi* (orange and green groups, respectively) by competitive enzyme-linked immunosorbent assay (cELISA) in the three-race modalities (*x*-axis). The horizontal line within each box represents the median race position of each group. Top and bottom box boundaries limit the interquartile range (IQR) between the first and third quartiles (25−75% of positions), and therefore boxes represent midspread (50% of less extreme positions). Whiskers show confidence limits based in the range of cases within 1.5 times the IQR. The dots show individual arrival positions according (horizontal jitter within groups is only to avoid dot overlapping). The colour of dots represents the percentage of inhibition PI in cELISA according to the colour scale wheaten red (high PI) and blue (low PI).

**Figure 2 animals-12-00637-f002:**
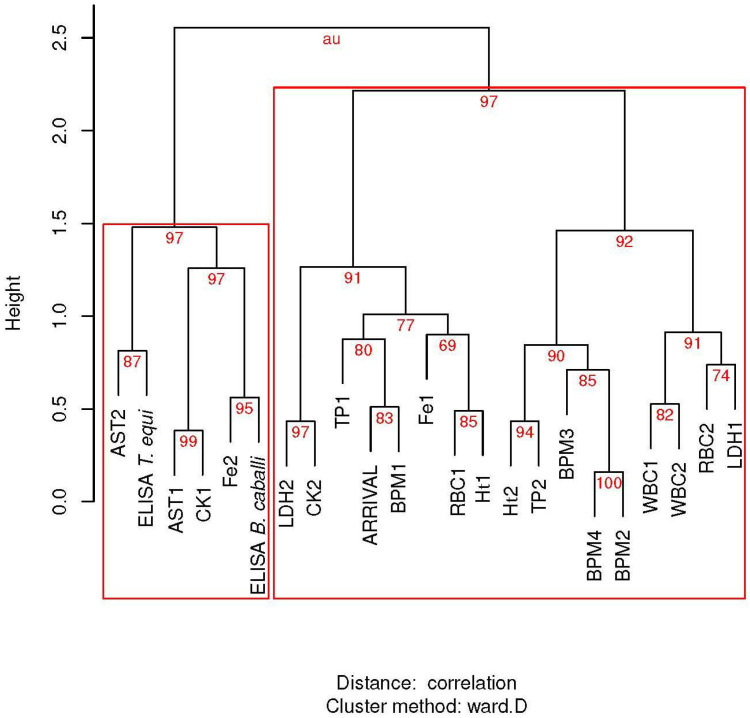
Hierarchical clustering via multiscale bootstrap resampling (n = 10,000) of arrival position in endurance horses with hematological and biochemical variables before and after the race, and with BPM controls (BPM1 to BPM4) during the competition. Repeated measures are indicated by the number of suffixes. Red numbers are the percentage of the signification of each cluster based on approximately unbiased *p*-values (au). Red rectangles show major significant clusters. AST: aspartate aminotransferase; CK: creatinine kinase; Fe: Iron; Ht: hematocrit value; LDH: lactate dehydrogenase; RBC: red blood cells; TP: total proteins; WBC: white blood cells.

**Figure 3 animals-12-00637-f003:**
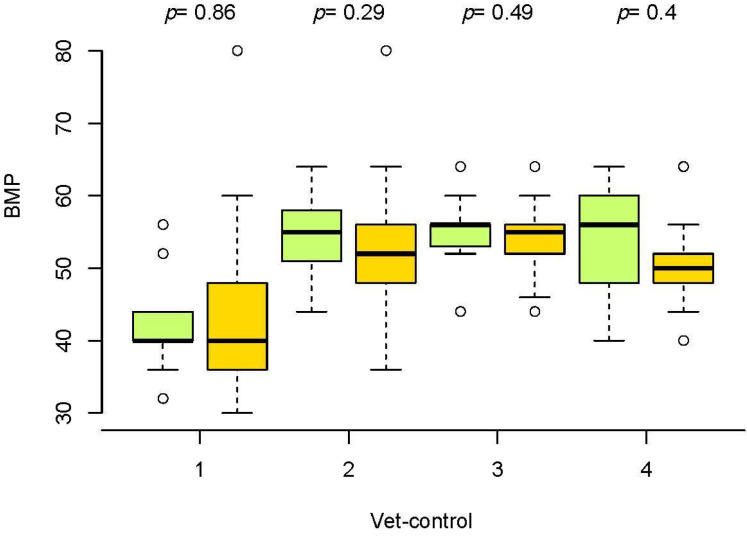
Box and whiskers plot of beats per minute (BPM) in veterinary controls (vet-control) before the race (1) and in vet-gates (2–4) in seronegative horses (green boxes) and ELISA-positive horses to EP (orange boxes).

**Figure 4 animals-12-00637-f004:**
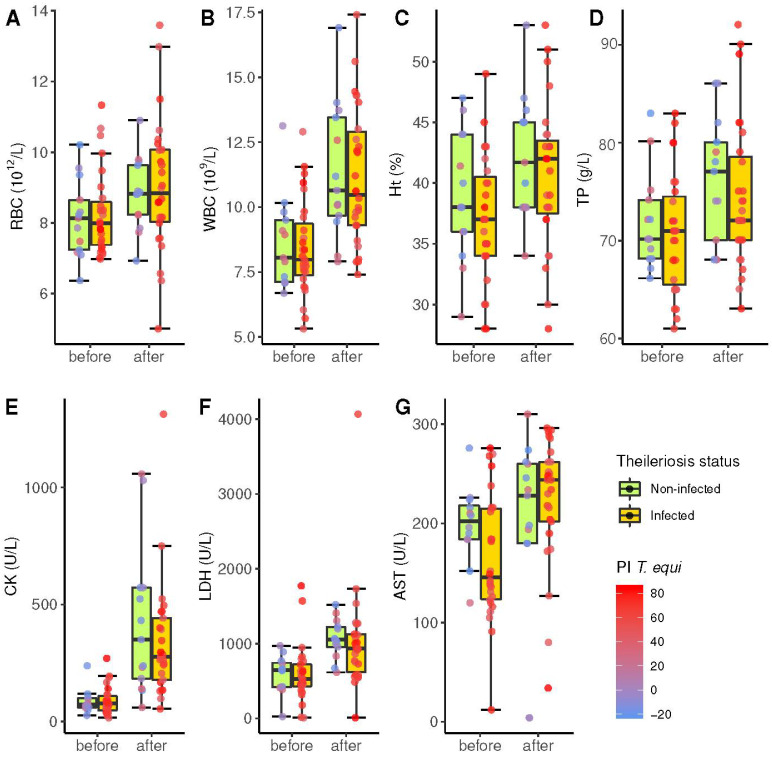
Box and whiskers plot of several hematological and biochemical parameters (*y*-axis) before and after the endurance challenge (*x*-axis) in 40 horses according to the diagnosis of theileriosis by competitive enzyme-linked immunosorbent assay (cELISA) (orange and green groups, respectively). The horizontal line within each box represents the median of each parameter. Top and bottom box boundaries limit the interquartile range (IQR) between the first and third quartiles. Whiskers show the range of cases within 1.5 times the IQR. The colour of dots represents the percentage of inhibition PI in cELISA according to the colour scale wheaten red (high PI) and blue (low PI). (**A**) RBC: red blood cells; (**B**) WBC: white blood cells; (**C**) Ht: hematocrit value; (**D**) TP: total pro-teins; (**E**) CK: creatinine kinase; (**F**) LDH: lactate de-hydrogenase; (**G**) AST: aspartate ami-notransferase.

**Table 1 animals-12-00637-t001:** Summary of mean ± standard deviation (SD) of hematologic and biochemistry parameters values depending on the presence of equine piroplasmosis (EP) and performance of the exercise.

				Global Group (n = 40)			Seropositive EP (n = 28)		Seronegative EP (n = 12)		
Parameter (SI Units)	Reference Values	n1	n2	Before Exercise	After Exercise	*p*-Value	Before Exercise	After Exercise	Before Exercise	After Exercise	*p*-Value
RBC (10^12^/L)	6.40−10.40	40	39	8.2 ± 1.1	9.0 ± 1.7	0.0046	8.2 ± 1.2	9.0 ± 1.9	8.3 ± 1.2	9.0 ± 1.9	0.41
Ht (%)	30−47	40	40	37.7 ± 5.3	41.2 ± 5.7	4.8 × 10^−4^	37.1 ± 5.2	40.6 ± 5.9	38.9 ± 5.2	42.6 ± 5.9	0.12
WBC (10^9^/L)	4.90−11.10	40	39	8.5 ± 1.8	10.8 ± 2.6	9.2 × 10^−8^	8.4 ± 1.8	10.8 ± 2.6	8.7 ± 1.8	11.0 ± 2.6	0.34
TP (g/L)	56−79	40	40	71 ± 6	75 ± 7	37 × 10^−4^	71 ± 6	74 ± 8	70 ± 6	77 ± 8	0.08
Iron (µmol/L)	18.8−49.6	38	37	28.5 ± 4.5	29.4 ± 10.2	3.7 × 10^−4^	28.1 ± 4.1	30.1 ± 11.8	29.5 ± 4.1	27.6 ± 11.8	0.68
CK (U/L)	10−350	38	38	88 ± 57	371 ± 281	1.8 × 10^−10^	90 ± 59	350 ± 256	85 ± 59	417 ± 256	0.33
LDH (U/L)	250−2070	37	39	592 ± 354	1039 ± 601	3.6 × 10^−5^	575 ± 388	1027 ± 704	600 ± 388	1066 ± 704	0.25
AST (U/L)	100−600	38	38	176 ± 59	214 ± 76	0.0018	167 ± 64	224 ± 65	197 ± 64	193 ± 65	0.36

*p*-value comparing the average of both measures in each animal. n1 = animals evaluated before exercise; n2 = animals evaluated after exercise; EP: equine piroplasmosis; RBC: red blood cells; Ht: hematocrit value; WBC: white blood cells; TP: total proteins; CK: creatinine kinase; LDH: lactate dehydrogenase; AST: aspartate aminotransferase.

## Data Availability

The data presented in this study are available on request from the corresponding author.

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
