# Peer review of "Effects of Competitive ELISA-Positive Results of Piroplasmosis on the Performance of Endurance Horses"

_animals, 2022, doi:10.3390/ani12050637_

Round 1

Reviewer 1 Report

In their manuscript titled "Effects of competitive ELISA positive against piroplasmosis on 2 performance in endurance horses", Bravo-Barriga et al. describe results from a serological study of racehorses suspected of being infected with pathogens that cause Equine piroplasmosis (EP). These pathogens cause a variety of physiological effects on the host that can impair race performance. The authors employed a competitive ELISA to screen for antibodies indicative of past infection. They also evaluated a variety of health parameters in race horses during race events. A large proportion of the horses had signs of sub-clinical infections (detected by cELISA) that did not appear to affect other physiological parameters or race performance. These finding indicate that sub-clinical infections should not disqualify horses from endurance competitions.

The manuscript is very clear and well written. There are only a few minor corrections required for clarity:

Line 47 - states "elimination of these competitions". That implies the events are canceled. I think the authors mean to say "elimination of the competitors"

Line 142 - What is meant by "and classified the competition," ? This is unclear.

Figure 1 - It is unclear why there are dots indicating both high and low cELISA values in the boxes for both negative and positive categories. Shouldn't the negative category be populated with low cELISA dots and the positive category be populated with high cELISA dots? The boxes are supposed to reflect the same cELISA results as the dots show. Please explain.

Fig. 4 - As for Fig 1, there is a confusing combination of red and blue dots in both infection status categories. In addition, "RAID" (shown in fig caption) is undefined. Please define.

Line 225 - Revise sentence. I think authors mean to say: "The only other seroprevalence study for EP in endurance horses has  been reported in Brazil by [28] with 77.3% participants in line with our findings."

Author Response

Authors: We appreciate the effort and work done to improve the article with the comments and suggestions provided.

We hope that these changes along with the responses to your following comments will be to your liking.

The manuscript is very clear and well written. There are only a few minor corrections required for clarity:

Line 47 - states "elimination of these competitions". That implies the events are canceled. I think the authors mean to say "elimination of the competitors"

Authors: It has been changed.

Line 142 - What is meant by "and classified the competition," ? This is unclear.

Authors: FEI and National regulations define classified the competition as: “Only Combinations in which the Horse has passed all the Horse Inspections at all stages of the Competition are entitled to be included in the final classification.”

There are 4 different situations not to classified:

  1. Failure to Qualify for the next Phase or for final classification occurs when a Combination is removed from the Competition for failure to pass a Horse Inspection, complete the full course as required, comply with applicable speed restrictions, and / or meet all time requirements for completion, or as a result of such other 'FTQ' designations as specified in Annex 3 (elimination codes).
  2. Disqualification occurs when a Combination is removed from a Competition and/or Event (or its results are subsequently disqualified after the Competition and/or Event) for a violation of these Endurance Rules or other FEI Rules and Regulations or the Competition Schedule.
  3. Withdrawal occurs when an Athlete withdraws their Horse from the Competition (without otherwise being removed from the Competition) at or prior to the First (Pre-Ride) Inspection.
  4. Retirement occurs when an Athlete decides (without otherwise being removed from the Competition) not to continue in the Competition after their Horse passes the First (Pre-Ride) Inspection, provided that this happens (i) before the Combination crosses the start line, or (ii) at the end of a Phase provided that the Combination has successfully completed that and any previous Phases and passes all Horse Inspections after each of those Phases, including any compulsory re-inspection or vet-requested re-inspection(the Horse must be considered fit to continue in the Competition at each of those inspections).

Figure 1 - It is unclear why there are dots indicating both high and low cELISA values in the boxes for both negative and positive categories. Shouldn't the negative category be populated with low cELISA dots, and the positive category be populated with high cELISA dots? The boxes are supposed to reflect the same cELISA results as the dots show. Please explain.

Authors: The legend of both figures has been improved for a better understanding by the reader.

Fig. 4 - As for Fig 1, there is a confusing combination of red and blue dots in both infection status categories. In addition, "RAID" (shown in fig caption) is undefined. Please define.

Authors: The explanation is the same as the previous one. The word RAID has been changed by endurance.

Line 225 - Revise sentence. I think authors mean to say: "The only other seroprevalence study for EP in endurance horses has been reported in Brazil by [28] with 77.3% participants in line with our findings."

Authors: The sentence have been rewrite.

Reviewer 2 Report

The submitted manuscript is a prospective study aimed to assess the impact of piroplasmosis on the athletic performance of endurance horses, and on some blood parameters, and whether these could be related to the elimination of horses during competition.

This reviewer considers the submitted manuscript as in general a good paper with a good sample population and fair study design. The results of the study would be of good interest to readers of the journal animals. However, there are a number of points to address before the manuscript can be recommended for publication.

The major point is that authors define the diagnosis of Theileria equi /Babesia caballi infection in horses based on the use of an indirect antibody detection test, the competitive enzyme-linked immunosorbent assay (cELISA). While this assay is listed by the World Health Organization as one of the available diagnostic tests for equine piroplasmosis, its fundament is based on the recognition of antibodies against the piroplasms that are produced in a susceptible horse after natural exposure to tick-transmitted Theileria equi (and/or Babesia caballi) parasites (although it can also be iatrogenically transmitted by, say, blood transfusions). However, a serologically positive animal does not necessarily mean that an animal is infected at that particular time when the blood sample was taken, particularly in clinically apparently uninfected animals, such as those selected for testing in this study. Other direct methods for diagnosis are recommended to detect truly infected horses such as the molecularly based assays (PCR, nPCR, rtPCR), specially indicated to detect persistent piroplasm infections, in those animals that resolved the infection either naturally or when the infection is cleared after treatment with, for example, Imidocarb dipropionate. It would be appropriate to include a paragraph indicating or justifying the reason(s) why a direct method of detection, other than Giemsa-stained blood smears, with higher analytical sensitivity to detect Piroplasm DNA, was not included in the study to ascertain that the horses selected for the study where individuals with persistent Theileria/Babesia infections, and not merely horses presenting with measurable residual antibodies produced long after previous exposure to piroplasm infection and/or after parasite clearance. Despite apparent clearance of T. equi, the cELISA can remain positive for up to 24 months in some horses (see for example Ueti et al, 2012).

Ueti MW, Mealey RH, Kappmeyer LS, White SN, Kumpula-McWhirter N, et al. (2012) Re-Emergence of the Apicomplexan Theileria equi in the United States: Elimination of Persistent Infection and Transmission Risk. PLoS ONE 7(9): e44713.

Author Response

Authors: We appreciate the time and work done by the reviewer to improve the article with the comments and suggestions provided.

We hope that the responses to your following comments will be to your liking.

The major point is that authors define the diagnosis of Theileria equi /Babesia caballi infection in horses based on the use of an indirect antibody detection test, the competitive enzyme-linked immunosorbent assay (cELISA). While this assay is listed by the World Health Organization as one of the available diagnostic tests for equine piroplasmosis, its fundament is based on the recognition of antibodies against the piroplasms that are produced in a susceptible horse after natural exposure to tick-transmitted Theileria equi(and/or Babesia caballi) parasites (although it can also be iatrogenically transmitted by, say, blood transfusions). However, a serologically positive animal does not necessarily mean that an animal is infected at that particular time when the blood sample was taken, particularly in clinically apparently uninfected animals, such as those selected for testing in this study. Other direct methods for diagnosis are recommended to detect truly infected horses such as the molecularly based assays (PCR, nPCR, rtPCR), specially indicated to detect persistent piroplasm infections, in those animals that resolved the infection either naturally or when the infection is cleared after treatment with, for example, Imidocarb dipropionate. It would be appropriate to include a paragraph indicating or justifying the reason(s) why a direct method of detection, other than Giemsa-stained blood smears, with higher analytical sensitivity to detect Piroplasm DNA, was not included in the study to ascertain that the horses selected for the study where individuals with persistent Theileria/Babesia infections, and not merely horses presenting with measurable residual antibodies produced long after previous exposure to piroplasm infection and/or after parasite clearance. Despite apparent clearance of T. equi, the cELISA can remain positive for up to 24 months in some horses (see for example Ueti et al, 2012).

Ueti MW, Mealey RH, Kappmeyer LS, White SN, Kumpula-McWhirter N, et al. (2012) Re-Emergence of the Apicomplexan Theileria equi in the United States: Elimination of Persistent Infection and Transmission Risk. PLoS ONE 7(9): e44713.

Authors:

We included only serological test, because the cELISA, is the regulatory test approved by the OIE for international horse transport, is considered to be the most sensitive test for chronic or inapparent T. equi infection. Definitive diagnosis of infection is most often accomplished with serologic testing performed.

PCR relies on the amplification and detection of parasite DNA isolated from the peripheral blood of an infected horse (if it is cantonated there may be diagnostic problems). It is an exquisitely sensitive test that when performed as a nested PCR can detect a positive result in an animal with T. equi parasitaemia low. However, the genetic variation reported between isolates of T. equi make the use of this test on a global scale challenging.

The PCR techniques have been not included in the study because only horses with infection shows PCR positives and it is not possible compete if the horse shows clinical signs. The current problem with the endurance horses is that if asymptomatic carriers can compete and how affect be positive in their performance. In asymptomatic horses the PCR is usually negative as have been demonstrated in several articles:

-           Lobanov VA, Peckle M, Massard CL, Brad Scandrett W, Gajadhar AA. Development and validation of a duplex real-time PCR assay for the diagnosis of equine piroplasmosis. Parasit Vectors. 2018 Mar 2;11(1):125.

“The duplex qPCR described here performed comparably to the existing single-target qPCR assays for T. equi and B. caballi and will be more cost-effective in terms of results turnaround time and reagent costs when both pathogens are being targeted for disease control and epidemiological investigations. These validation data also support the reliability of the ema-1 gene-specific oligonucleotides developed in this study for confirmatory testing of non-negative serological test results for T. equi by qPCR.

However, the B. caballi-specific qPCR cannot be similarly recommended as a confirmatory assay for routine regulatory testing due to the low level of agreement with serological test results demonstrated in this study. Further studies are needed to determine the transmission risk posed by PCR-negative equines with detectable antibodies to B. caballi.”

  1. Lobanov VA, Peckle M, Massard CL, Brad Scandrett W, Gajadhar AA. Development and validation of a duplex real-time PCR assay for the diagnosis of equine piroplasmosis. Parasit Vectors. 2018 Mar 2;11(1):125.

“Competitive enzyme-linked immunosorbent assay (cELISA) test appears to be more reliable than microscopic examination and PCR in estimating the seroprevalence of the disease as well as identifying carrier horses to babesiosis.”

  1. Coultous RM, Phipps P, Dalley C, Lewis J, Hammond TA, Shiels BR, Weir W, Sutton DGM. Equine piroplasmosis status in the UK: an assessment of laboratory diagnostic submissions and techniques. Vet Rec. 2019 Jan 19;184(3):95.

“Equine piroplasmosis (EP) has historically been of minor concern to UK equine practitioners, primarily due to a lack of competent tick vectors. However, increased detection of EP tick vector species in the UK has been reported recently. EP screening is not currently required for equine importation, and when combined with recent relaxations in movement regulations, there is an increased risk regarding disease incursion and establishment into the UK. This study evaluated the prevalence of EP by both serology and PCR among 1242 UK equine samples submitted for EP screening between February and December 2016 to the Animal and Plant Health Agency and the Animal Health Trust. Where information was available, 81.5 per cent of submissions were for the purpose of UK export testing, and less than 0.1 per cent for UK importation. Serological prevalence of EP was 8.0 per cent, and parasite DNA was found in 0.8 per cent of samples. A subsequent analysis of PCR sensitivity in archived clinical samples indicated that the proportion of PCR-positive animals is likely to be considerably higher. The authors conclude that the current threat imposed by UK carrier horses is not adequately monitored and further measures are required to improve national biosecurity and prevent endemic disease.”

Despite these limitations of molecular techniques in EP, we agree with the reviewer that their use could have been a complement in diagnosis.

However, the authors decided to use the assay (cELISA), for the above reasons, and, as the reviewer has indicated, because it is listed by the World Health Organization as one of the available diagnostic tests for equine piroplasmosis.

Our study aims to be a first approach to the problem of EP in horse competitions. We hope that this preliminary study indicates the need to continue in this line in the future both for other groups and for us. We hope that we can advance with larger studies, and we will include the techniques (PCR, nPCR, rtPCR) recommended by the reviewer for comparative analyses.

Following the reviewer's recommendation, we have placed a text indicating this possible limitation (lines 341-345).

Reviewer 3 Report

The aim of the study was to determine the effects of competitive ELISA positive against piroplasmosis on performance in endurance horses.

The approach of the study appears original. And it is of great interest for the professional sport training. Especially for trainers, owners and veterinary practitioners but also for the enthusiasts. The present paper is interesting. However, I found some parts which should be corrected.

Introduction

L43: please add what parameters are examinated (heart rate, mucous membranes (colour and moisture), dehydration (measured as the time it takes for a pinched skin fold over the point of the shoulder to flatten), gut sounds, muscle condition and regularity of gait (evaluated in trot).

L47: authors write about various diagnostic techniques but more explanation is needed. Authors omitted the information about changes of serum Amyloid A (SAA) concentration during endurance training which seems to be good health indicator of the horse. Also there are changes in cytokines concentration which leads to creation the anti-inflammatory state during endurance training. Thus, they seems to be the biomarkes in the future. In addition, the most novel performance monitoring techniques are used in race horses. For example changes in PBMCs proliferation and activity, or cytokines mRNA expression are implemented to training monitoring.

Materials&methods

Please add the information about age, breed, sex of the examined horses.

L87 – add EDTA

L88 – add information about time of the blood sampling. Was it similar in each horse?

L92 – how long blood samples waited until processing

L114 – use Ht not HV, WBC is from white blood cell count

L115 – Was everything calibrated for equine species?

Results

Table 1 add the column with references values for equine species

Discussion

Discussion is brief and well written. Easy to understand for the reader.

L229 – ad information why PCR was not conducted as it is the limitation of the study.

L239-250 – AST in this case mostly is origin from the muscle tissue. CPK and AST activities are commonly used in horses as the indicators of any kind of muscle fatigue or damage. After strenuous exercise, they increase from four- to 35-fold and from two- to six-fold, respectively. At short distances (+/- 40 km), CPK and AST activities are elevated significantly but only slightly less than two-fold, regardless of the type of effort which was confirmed in very recent studies.

L265 – In some studies it was postulated that the increase in resting RBC, HT, HGB is more likely to be seen in horses that have had a significant period of detraining or rest prior to starting exercise. However, the most important is the spleen reserve for endurance horses.

L278 – please add mean values od WBC, however it was without statistical significance.

L283 – I postulate that dehydration in this case is connected with the distance. Thus, Authors should take into account the distance and haematological changes.

References

Authors shoud discuss their results with more recent publications.

Author Response

The aim of the study was to determine the effects of competitive ELISA positive against piroplasmosis on performance in endurance horses.

The approach of the study appears original. And it is of great interest for the professional sport training. Especially for trainers, owners and veterinary practitioners but also for the enthusiasts. The present paper is interesting. However, I found some parts which should be corrected.

Authors: We appreciate the time and work done by the reviewer to improve the article with the comments and suggestions provided.

We hope that the responses to your following comments will be to your liking.

Introduction

L43: please add what parameters are examinated (heart rate, mucous membranes (colour and moisture), dehydration (measured as the time it takes for a pinched skin fold over the point of the shoulder to flatten), gut sounds, muscle condition and regularity of gait (evaluated in trot).

Authors: It have been added.

L47: authors write about various diagnostic techniques but more explanation is needed. Authors omitted the information about changes of serum Amyloid A (SAA) concentration during endurance training which seems to be good health indicator of the horse. Also there are changes in cytokines concentration which leads to creation the anti-inflammatory state during endurance training. Thus, they seems to be the biomarkes in the future. In addition, the most novel performance monitoring techniques are used in race horses. For example changes in PBMCs proliferation and activity, or cytokines mRNA expression are implemented to training monitoring.

It have been demonstrated different changes in the cytokine profiles and in other acute phase proteins during high performing sports. SAA is a good indicator of good health but is very unspecify and in the previous articles the results showed that is useless in race training (Cywinska et al. 2013). The article from 2010 using SAA as indicator of status included horses in competitions of 120 and 160 kms (Cywinska et al. 2010).. We included only competitions until 80 kms and the results probably were more similar at training than a maximum level of competition. The aim of the study was only evaluate the impact of subclinical piroplasmosis in the endurance performance, but we can consider include other parameters in future researches.

Cywinska A, Witkowski L, Szarska E, Schollenberger A, Winnicka A. Serum amyloid A (SAA) concentration after training sessions in Arabian race and endurance horses. BMC Vet Res. 2013 May 1;9:91.

Cywinska A, Gorecka R, Szarska E, Witkowski L, Dziekan P, Schollenberger A. Serum amyloid A level as a potential indicator of the status of endurance horses. Equine Vet J Suppl. 2010 Nov;(38):23-7.

Materials&methods

Please add the information about age, breed, sex of the examined horses.

L87 – add EDTA

Authors: Done.

L88 – add information about time of the blood sampling. Was it similar in each horse?

Authors: Some extra explanation has been added on lines 95-97. All the horses included in the study were sampling in similar conditions:

1- First sample was taken after the first clinical examination in the vet gate, previous the competition. Not all the horses are evaluated at the same time, each one has a number and depends on the number of vets and the order of entry in the competition venue.

2- Second sample was obtained at the end of the competition or when horses were eliminated, just after finishing the veterinary control, previous to send the horse to his stall.

L92 – how long blood samples waited until processing

Authors:  Heparin samples were centrifugated and stored immediately. The EDTA samples were analyzed after the competition, the same day that were obtained, 6 hours for the first sample and 1 hour for the second sample.

L114 – use Ht not HV, WBC is from white blood cell count

Authors: It has been changed

L115 – Was everything calibrated for equine species?

Authors: All the equipment and techniques have been validated for equids previously.

Results

Table 1 add the column with references values for equine species

Authors: Done. That information has been added in a new column.

Discussion

Discussion is brief and well written. Easy to understand for the reader.

L229 – ad information why PCR was not conducted as it is the limitation of the study.

Author:

We included only serological test, because the cELISA, is the regulatory test approved by the OIE for international horse transport, is considered to be the most sensitive test for chronic or inapparent T. equi infection. Definitive diagnosis of infection is most often accomplished with serologic testing performed.

PCR relies on the amplification and detection of parasite DNA isolated from the peripheral blood of an infected horse (if it is cantonated there may be diagnostic problems). It is an exquisitely sensitive test that when performed as a nested PCR can detect a positive result in an animal with T. equi parasitaemia low. However, the genetic variation reported between isolates of T. equi make the use of this test on a global scale challenging.

The PCR techniques have been not included in the study because only horses with infection shows PCR positives and it is not possible compete if the horse shows clinical signs. The current problem with the endurance horses is that if asymptomatic carriers can compete and how affect be positive in their performance. In asymptomatic horses the PCR is usually negative as have been demonstrated in several articles:

-           Lobanov VA, Peckle M, Massard CL, Brad Scandrett W, Gajadhar AA. Development and validation of a duplex real-time PCR assay for the diagnosis of equine piroplasmosis. Parasit Vectors. 2018 Mar 2;11(1):125.

“The duplex qPCR described here performed comparably to the existing single-target qPCR assays for T. equi and B. caballi and will be more cost-effective in terms of results turnaround time and reagent costs when both pathogens are being targeted for disease control and epidemiological investigations. These validation data also support the reliability of the ema-1 gene-specific oligonucleotides developed in this study for confirmatory testing of non-negative serological test results for T. equi by qPCR.

However, the B. caballi-specific qPCR cannot be similarly recommended as a confirmatory assay for routine regulatory testing due to the low level of agreement with serological test results demonstrated in this study. Further studies are needed to determine the transmission risk posed by PCR-negative equines with detectable antibodies to B. caballi.”

  1. Lobanov VA, Peckle M, Massard CL, Brad Scandrett W, Gajadhar AA. Development and validation of a duplex real-time PCR assay for the diagnosis of equine piroplasmosis. Parasit Vectors. 2018 Mar 2;11(1):125.

“Competitive enzyme-linked immunosorbent assay (cELISA) test appears to be more reliable than microscopic examination and PCR in estimating the seroprevalence of the disease as well as identifying carrier horses to babesiosis.”

  1. Coultous RM, Phipps P, Dalley C, Lewis J, Hammond TA, Shiels BR, Weir W, Sutton DGM. Equine piroplasmosis status in the UK: an assessment of laboratory diagnostic submissions and techniques. Vet Rec. 2019 Jan 19;184(3):95.

“Equine piroplasmosis (EP) has historically been of minor concern to UK equine practitioners, primarily due to a lack of competent tick vectors. However, increased detection of EP tick vector species in the UK has been reported recently. EP screening is not currently required for equine importation, and when combined with recent relaxations in movement regulations, there is an increased risk regarding disease incursion and establishment into the UK. This study evaluated the prevalence of EP by both serology and PCR among 1242 UK equine samples submitted for EP screening between February and December 2016 to the Animal and Plant Health Agency and the Animal Health Trust. Where information was available, 81.5 per cent of submissions were for the purpose of UK export testing, and less than 0.1 per cent for UK importation. Serological prevalence of EP was 8.0 per cent, and parasite DNA was found in 0.8 per cent of samples. A subsequent analysis of PCR sensitivity in archived clinical samples indicated that the proportion of PCR-positive animals is likely to be considerably higher. The authors conclude that the current threat imposed by UK carrier horses is not adequately monitored and further measures are required to improve national biosecurity and prevent endemic disease.”

Despite these limitations of molecular techniques in EP, we agree with the reviewer that their use could have been a complement in diagnosis.

However, the authors decided to use the assay (cELISA), for the above reasons, and, as the reviewer has indicated, because it is listed by the World Health Organization as one of the available diagnostic tests for equine piroplasmosis.

Our study aims to be a first approach to the problem of EP in horse competitions. We hope that this preliminary study indicates the need to continue in this line in the future both for other groups and for us. We hope that we can advance with larger studies, and we will include the techniques (PCR, nPCR, rtPCR) recommended by the reviewer for comparative analyses.

Following the reviewer's recommendation, we have placed a text indicating this possible limitation (lines 341-345).

L239-250 – AST in this case mostly is origin from the muscle tissue. CPK and AST activities are commonly used in horses as the indicators of any kind of muscle fatigue or damage. After strenuous exercise, they increase from four- to 35-fold and from two- to six-fold, respectively. At short distances (+/- 40 km), CPK and AST activities are elevated significantly but only slightly less than two-fold, regardless of the type of effort which was confirmed in very recent studies.

Authors: We included a reference.

L265 – In some studies it was postulated that the increase in resting RBC, HT, HGB is more likely to be seen in horses that have had a significant period of detraining or rest prior to starting exercise. However, the most important is the spleen reserve for endurance horses.

Authors: This is true but, in the discussion, we were trying to interpreter the results comparing seropositive and seronegative horses. The discussion is focus on why seropositive animals had lower RBC count than seronegative horses. Moreover, the explanation about the spleen role is included between lines 272 to 274.

L278 – please add mean values od WBC, however it was without statistical significance.

Authors: It has been added

L283 – I postulate that dehydration in this case is connected with the distance. Thus, Authors should take into account the distance and haematological changes.

Authors: It have been taken account and also the other possibilities of increased total proteins after race. It is difficult interpreter the degree of dehydration and compare between group because the Ht in the serologically positive horses were lower than the serologically negative group.

References

Authors should discuss their results with more recent publications.

Authors: We added some new references.

  • Witkowska-Piłaszewicz O, Bąska P, Czopowicz M, Żmigrodzka M, Szczepaniak J, Szarska E, Winnicka A, Cywińska A. Changes in Serum Amyloid A (SAA) Concentration in Arabian Endurance Horses During First Training Season. Animals. 2019 Jun 8;9(6):330.
  • Maśko M, Domino M, Jasiński T, Witkowska-Piłaszewicz O.The Physical Activity-Dependent Hematological and Biochemical Changes in School Horses in Comparison to Blood Profiles in Endurance and Race Horses. Animals. 2021 Apr 14;11(4):1128.

Round 2

Reviewer 2 Report

The submitted manuscript is a prospective study aimed to assess the impact of piroplasmosis on the athletic performance of endurance horses as well as on some blood parameters, and whether these could be related to the elimination of horses during competition.

This reviewer considers the submitted manuscript as in general a good paper with a good sample population and fair study design. The results of the study would be of good interest to readers of the journal animals. Authors have determined appropriate to include a paragraph indicating that a PCR-based detection technique was not performed in this study, so that some positive animals may have escaped diagnosis, recommending to use molecular diagnostic techniques in future studies as a complement to immunological assays.

Author Response

Thank you for the effort to improve the article.

Reviewer 3 Report

Now all my concerns were explained. Now I think that article is ready for publication.

Author Response

Thank you to improve our article.